# Efficient Computation of Signature-Restricted Views for Semantic Web Ontologies

## ABSTRACT

Uniform Interpolation (UI) is an advanced reasoning procedure used to narrow down an ontology to a restricted view. This new ontology, known as a uniform interpolant, will only consist of the "relevant names", yet it will retain their original meanings. UI is immensely promising due to its applicability across various domains where custom views of ontologies are essential. Nonetheless, to unlock its full potential, we need optimized techniques to generate these tailored views. Previous studies suggest that creating uniform interpolants for $\mathcal{EL}$-ontologies is notably challenging. In some instances, it is not even feasible to compute a uniform interpolant. When feasible, the size of the uniform interpolant can be up to triple exponentially larger than the source ontology. Despite these challenges, our paper introduces an improved "forgetting" technique specifically designed for computing uniform interpolants of $\mathcal{ELI}$-ontologies. We demonstrate that, with good normalization and inference strategies, such uniform interpolants can be efficiently computed, just as swiftly as computing "modules". A comprehensive evaluation with a prototypical implementation of the method shows superb success rates over two popular benchmark datasets, demonstrating a clear computational advantage over state-of-the-art approaches.

## CCS CONCEPTS

• **Theory of computation → Description logics**; **Automated reasoning**; • **Computing methodologies → Ontology engineering**.

## KEYWORDS

Ontologies, Module Extraction, Uniform Interpolation, Forgetting

**ACM Reference Format:**
Anonymous Author(s). 2018. Efficient Computation of Signature-Restricted Views for Semantic Web Ontologies. In *Proceedings of Make sure to enter the correct conference title from your rights confirmation emai (Conference acronym 'XX)*. ACM, New York, NY, USA, 9 pages. https://doi.org/XXXXXXX.XXXXXXX

## 1 INTRODUCTION

The increasing availability of machine-processable web data has put the desideratum of semantic interoperability on the top of the World Wide Web's agenda — a requirement to enable the exchange of data with precise, unambiguous, shared meaning across distributed web applications. Although it is not yet a reality in the Web of today, much effort and progress have been made towards achieving this vision — *the Semantic Web* [39]. The key idea is to add descriptions about the web data (aka metadata), linking each data element to a controlled vocabulary that provides a common reference point for aggregating and comparing data about a particular subject domain. These descriptions can rely on logical statements relating data to some terms within a given ontology [12, 40].

Ontologies fix a controlled vocabulary of names (aka *signature*) relevant to a subject domain and specify constraints among the names by logical statements (aka *axioms*) [40]. However, due to the intrinsic heterogeneity of web resources, ontologies designed for the semantic web tend to exhibit large-scale and encompass knowledge spanning a broad spectrum of topics. Nonetheless, this sheer scale and comprehensiveness may hinder the reusability of ontologies in real-world web applications. This is primarily attributed to the challenges associated with the management and manipulation of large and complex ontologies, which can be unwieldy and pose considerable computational costs when engaged in the reasoning process. One possible strategy to address these challenges is to extract a "module" from an ontology that retains the functionality of the original ontology within a specific context while achieving a substantial reduction in size. This is a desirable strategy for the flexible reuse of ontologies for several reasons. First, many ontology maintenance tasks can be done locally by simply adjusting the specific module in question. Next, the single components or respective modules can be reused in other contexts more easily. Further, from a more technical perspective, reasoning tasks can be done more efficiently under certain circumstances, as only a small module might be relevant for specific deductions, or the reasoning itself can be distributed to several machines separately handling the modules. A general module $\mathcal{M}$ is defined as [10]:

**Condition I:** a syntactic subset of a given ontology $O$;
**Condition II:** preserves all logical entailments w.r.t. a specific subsignature $\Sigma$ of $O$.

This means that $\mathcal{M}$ and $O$ align in their logical entailments w.r.t. $\Sigma$, and $\mathcal{M}$ can therefore be reused in other contexts within $\Sigma$ as a substitute for $O$. To satisfy both of the above conditions, $\mathcal{M}$ often needs to incorporate names outside of $\Sigma$. Take, for example, $O = \{A \sqsubseteq B, B \sqsubseteq C, D \sqsubseteq E\}$ and $\Sigma = \{A, C\}$. A module $\mathcal{M}$ of $O$ w.r.t. $\Sigma$ is $\{A \sqsubseteq B, B \sqsubseteq C\}$. While this module preserves all logical entailments over $\{A, C\}$, it must include $B$ — a name not in $\Sigma$ — to achieve this preservation. Such constraints can limit the reusability of ontologies in specific real-world scenarios. For instance, in domains like medicine or the military, ontologies might hold sensitive data that should remain undisclosed when these ontologies are made public, distributed, or shared. This is also pertinent in industrial settings where proprietary details need stringent protection. A possible solution to ensure confidentiality is to limit the exposure of names considered sensitive. One strategy to manage this concealed information is to disseminate a fragment of the ontology that only includes names particular users have permission to view. This becomes especially vital when ontology proprietors wish to share their data with other users or the general public, but aim to disclose only non-sensitive details. Directly repurposing a module from an ontology may not be suitable in this context since it cannot assure the absolute concealment of specific names; the module might still include names that fall outside the designated signature.

In situations where one needs to preserve the functionality of the source ontology but only wishes to utilize a specific subset of names, it is more advantageous to have a signature-restricted view of the original ontology. This paper considers creating signature-restricted views using a uniform interpolation approach. Essentially, Uniform Interpolation (UI) is an advanced reasoning procedure that aims to narrow down an ontology to a smaller signature. When provided with an ontology that utilizes a specific sub-signature, $\Sigma$, representing the "relevant names" associated with certain topics, UI computes a new ontology, known as a *uniform interpolant*, that only employs the names in $\Sigma$ while maintaining the same semantics of the $\Sigma$-names. However, a uniform interpolant does not just take a part of the original ontology; it might include axioms not found in the original. Think of it as a condensed version of a module. To ensure the semantics of the $\Sigma$-names remain intact, numerous new axioms will be derived from the original ontology. As a result, uniform interpolants can contain substantially more axioms than the source ontology. In fact, research indicates that the UI process is more computationally challenging than modularization [4, 29, 43].

Nevertheless, creating signature-restricted views of ontologies can be of great importance since it may be used in a variety of applications where suitable views of ontologies need to be computed, such as debugging and repair [34, 42], merging and alignment [24, 33, 44], versioning [13, 14, 38], semantic difference [16, 17, 25, 49], abduction and explanation generation [6, 21] and interactive ontology revision [32]. However, this potential can only be fully realized if a highly optimized method (and its corresponding implementation) for computing such views exists.

In this paper, we present a highly optimized method for computing uniform interpolants of $\mathcal{ELI}$-ontologies. The method is based on a "forgetting procedure" that computes uniform interpolants by singly eliminating names from the original ontology that do not belong in $\Sigma$. Nikitina and Rudolph [31] show that computing uniform interpolants of $\mathcal{EL}$-ontologies is computationally extremely hard — a finite uniform interpolant does not always exist, and if it exists, then there exists one of at most triple exponential size in terms of the input ontology, and that, in the worst case, no shorter uniform interpolant exists. We show however in this paper that: (i) this result should not constitute a fundamental technical obstacle for UI in practice, and (ii) with good normalization and inference strategies, uniform interpolants can be computed as fast as computing modules. A comprehensive evaluation with a prototypical implementation of the method shows superb success rates over two popular benchmark datasets, demonstrating a clear computational advantage over state-of-the-art tools.

A long version of this paper including all missing proofs and additional illustrative examples, as well as the source code for the prototypical UI implementation alongside the test datasets, are *anonymously* distributed for review at https://github.com/anonymous-ai-researcher/www2024.

## 1.1 Related Work

*Forgetting* is an inherently difficult (non-standard) reasoning problem concerned with eliminating from an ontology a set of concept and role names in its signature, namely the *forgetting signature*, in such a way that all logical entailments are preserved up to the remaining signature; it is much harder than standard reasoning (satisfiability testing), and very few logics are known to be complete for forgetting. Foundational studies have shown that: (i) forgetting solutions do not always exist for the DL $\mathcal{EL}$ or $\mathcal{ALC}$ [15, 16, 29], (ii) deciding the existence of forgetting solutions is ExpTime-complete for $\mathcal{EL}$ [26] and 2ExpTime-complete for $\mathcal{ALC}$ [29], and (iii) forgetting solutions can be triple exponential in size w.r.t. the input ontologies for $\mathcal{EL}$ and $\mathcal{ALC}$ [29, 31].

Although forgetting is a challenging problem, there is however general consensus on its tremendous potential for ontology-based knowledge processing, and there have been continuous efforts dedicated to the development and automation of practical methods for computing solutions of forgetting. A few such methods have thus been developed and automated for various DLs.

Presently, the only available forgetting tools are Lethe and Fame. Lethe [19] utilizes the classic *resolution* inference system [3, 7], and handles ontologies specified in $\mathcal{ALC}$ and several extensions. Fame [50] considers a stronger notion of forgetting, namely model-theoretic forgetting [45]; the method is based on a monotonicity property known as *Ackermann's Lemma* [1], and accommodates ontologies as expressive as $\mathcal{ALCOIH}$. The tools of Nui and [47] are another two resolution-based approaches for $\mathcal{EL}$- and $\mathcal{SHQ}$-ontologies, respectively, but neither remains accessible at the moment. Thus, in this paper, our baselines are Lethe and Fame.

## 2 PRELIMINARIES

Let $N_C$ and $N_R$ be pairwise disjoint and countably infinite sets of *concept* and *role* names, respectively. *Roles* in $\mathcal{ELI}$ are a role name $r \in N_R$ or its inverse $r^-$. *Concept descriptions* (or *concepts* for short) in $\mathcal{ELI}$ have one of the following forms:

$$\top \mid A \mid C \sqcap D \mid \exists r.C \mid \exists r^-.C,$$

where $A \in N_C$, $r \in N_R$, and $C$ and $D$ range over concepts. We use $r^-$ to denote $s$ if $r = s^-$ for $s \in N_R$ and identify $(r^-)^-$ with r.

An $\mathcal{ELI}$-ontology $O$ is a finite set of *axioms* of the form $C \sqsubseteq D$ (*general concept inclusion*, or GCI), where $C$ and $D$ are concepts. We use $C \equiv D$ as an abbreviation for the GCIs $C \sqsubseteq D$ and $D \sqsubseteq C$.

Let $\mathcal{S} \in N_C \cup N_R$ be a designated concept name or role name. A concept (axiom) is called an $\mathcal{S}$-concept ($\mathcal{S}$-axiom) if it contains $\mathcal{S}$. An occurrence of $\mathcal{S}$ is said to be *positive* (*negative*) in an $\mathcal{S}$-axiom if it occurs at the right-hand (left-hand) side of the axiom.

The semantics of $\mathcal{ELI}$ is defined in terms of an *interpretation* $\mathcal{I} = \langle \Delta^{\mathcal{I}}, \cdot^{\mathcal{I}} \rangle$, where $\Delta^{\mathcal{I}}$ is a non-empty set, known as the *domain of the interpretation*, and $\cdot^{\mathcal{I}}$ is the *interpretation function* that maps every concept name $A \in N_C$ to a set $A^{\mathcal{I}} \subseteq \Delta^{\mathcal{I}}$, and every role name $r \in N_R$ to a binary relation $r^{\mathcal{I}} \subseteq \Delta^{\mathcal{I}} \times \Delta^{\mathcal{I}}$. The interpretation function $\cdot^{\mathcal{I}}$ is inductively extended to concepts as follows:

$$\top^{\mathcal{I}} = \Delta^{\mathcal{I}} \qquad (C \sqcap D)^{\mathcal{I}} = C^{\mathcal{I}} \cap D^{\mathcal{I}}$$

$$(\exists r.C)^{\mathcal{I}} = \{x \in \Delta^{\mathcal{I}} \mid \exists y.(x,y) \in r^{\mathcal{I}} \wedge y \in C^{\mathcal{I}}\}$$

$$(r^-)^{\mathcal{I}} = \{(y,x) \in \Delta^{\mathcal{I}} \times \Delta^{\mathcal{I}} \mid (x,y) \in r^{\mathcal{I}}\}$$

Let $\mathcal{I}$ be an interpretation. A GCI $C \sqsubseteq D$ is *true* in $\mathcal{I}$ iff $C^{\mathcal{I}} \subseteq D^{\mathcal{I}}$. $\mathcal{I}$ is a *model* of an ontology $O$, written $\mathcal{I} \models O$, iff every axiom in $O$ is *true* in $\mathcal{I}$. An axiom $\alpha$ is a *logical entailment* of $O$, written $O \models \alpha$, iff $\alpha$ is true in every model $\mathcal{I}$ of $O$.

A *signature* $\Sigma \subseteq \mathsf{N_C} \cup \mathsf{N_R}$ is a finite set of concept names and role names. We denote by $\mathsf{sig_C}(X)$ and $\mathsf{sig_R}(X)$ the sets of respectively the concept names and role names present in $X$, where $X$ can be any syntactic objects including concepts, roles, axioms, and ontologies. We further define $\mathsf{sig}(X) = \mathsf{sig_C}(X) \cup \mathsf{sig_R}(X)$.

**DEFINITION 1 (FORGETTING).** *Let $O$ be an $\mathcal{ELI}$-ontology and $S \in \mathsf{sig}(O)$ be a concept/role name, referred to as the* pivot. *An $\mathcal{ELI}$-ontology $\mathcal{V}$ is a result of forgetting $\{S\}$ from $O$ if the following conditions hold:*

- *(i) $\mathsf{sig}(\mathcal{V}) \subseteq \mathsf{sig}(O) \backslash \{S\}$, and*
- *(ii) for any $\mathcal{ELI}$-axiom $\alpha$ with $\mathsf{sig}(\alpha) \subseteq \mathsf{sig}(O) \backslash \{S\}$, $\mathcal{V} \models \alpha$ iff $O \models \alpha$.*

*More generally, let $\mathcal{F} \subseteq \mathsf{sig}(O)$ be a finite set of concept and role names, referred to as the* forgetting signature. *An $\mathcal{ELI}$-ontology $\mathcal{V}$ is a result of forgetting $\mathcal{F}$ from $O$ if the following conditions hold:*

- *(i) $\mathsf{sig}(\mathcal{V}) \subseteq \mathsf{sig}(O) \backslash \mathcal{F}$, and*
- *(ii) for any $\mathcal{ELI}$-axiom $\alpha$ with $\mathsf{sig}(\alpha) \subseteq \mathsf{sig}(O) \backslash \mathcal{F}$, $\mathcal{V} \models \alpha$ iff $O \models \alpha$.*

The process of "forgetting" distills an ontology $O$ into a more refined perspective, $\mathcal{V}$, which is anchored sorely on a sub-signature $\Sigma$ of $O$. Here, $\Sigma$ is defined as $\Sigma \subseteq \mathsf{sig}(O) \backslash \mathcal{F}$. Notably, when considering $\Sigma$, $\mathcal{V}$ mirrors the behavior of $O$ within the $\mathcal{ELI}$ framework, implying that both ontologies yield identical $\mathcal{ELI}$-entailments w.r.t. $\Sigma$. Forgetting can also be defined in terms of *inseparability* [4, 15, 28] and *conservative extensions* [9, 27]: an $\mathcal{ELI}$-ontology $\mathcal{V}$ is the result of forgetting $\mathcal{F}$ from an $\mathcal{ELI}$-ontology $O$, where $\mathcal{F} = \mathsf{sig}(O) \backslash \Sigma$ iff $\mathcal{V} \equiv_{\Sigma}^{\mathcal{ELI}} O$, and $O$ is an $\mathcal{ELI}$-conservative extension of $\mathcal{V}$ iff $\mathcal{V} \subseteq O$ and $\mathcal{V} \equiv_{\Sigma}^{\mathcal{ELI}} O$. The results of forgetting are unique up to logical equivalence, i.e., should both $\mathcal{V}_1$ and $\mathcal{V}_2$ result from the forgetting of $\mathcal{F}$ from $O$, they would be logically indistinguishable, though their manifest representations may differ.

## 3 NORMALIZATION OF $\mathcal{ELI}$-ONTOLOGIES

Our method computes the result of forgetting $\mathcal{F}$ from $O$ by iteratively forgetting single names in $\mathcal{F}$. The calculus for single-name elimination works on specialized normal forms of $\mathcal{ELI}$-ontologies.

### 3.1 A-Normal Form (A-NF)

**DEFINITION 2 (A-NORMAL FORM).** *We say that an axiom is in A-normal form (or A-NF for short) if it has one of the following forms, where (i) $r, s \in \mathsf{N_R}$, and (ii) B, C, D, E, and F are concepts that do not*

|  | *A-NF* |  | *A-NF* |
|---|---|---|---|
| *I* | $C \sqsubseteq A$ | *IV* | $A \sqcap E \sqsubseteq F$ |
| *II* | $C \sqsubseteq \exists r.(A \sqcap D)$ | *V* | $\exists s.(A \sqcap E) \sqcap F \sqsubseteq B$ |
| *III* | $C \sqsubseteq \exists r^-.(A \sqcap D)$ | *VI* | $\exists s^-.(A \sqcap E) \sqcap F \sqsubseteq B$ |

*contain A. An $\mathcal{ELI}$-ontology $O$ is in A-NF if every A-axiom in $O$ is in A-NF.*

One can transform a given $\mathcal{ELI}$ ontology $O$ into a normalized one by exhaustively applying the following normalization rules to the A-axioms in $O$ that have yet to be in A-NF ($X, Y, Y_1$ and $Y_2$ are $\mathcal{ELI}$-concepts).

(1) For each instance of a GCI $X \sqsubseteq Y_1 \sqcap Y_2$, if either $Y_1$ or $Y_2$ contains A, replace it by $X \sqsubseteq Y_1$ and $X \sqsubseteq Y_2$;

(2) For each instance of $X \sqsubseteq Y$, if A occurs more than once in it and a concept of the form $\exists R.C$ is present at the surface level of the left-hand side $X$, where $R$ is a role and $C$ is a concept that contains A, replace $C$ by a fresh definer $Z \in \mathsf{N_C}$ and add $C \sqsubseteq Z$ to $O$;

(3) For each instance of $X \sqsubseteq Y$, if A occurs more than once in it and a concept of the form $\exists R.C$ is present at the surface level of the right-hand side $Y$, where $R$ is a role and $C$ is a concept that contains A, replace $C$ by a fresh definer $Z \in \mathsf{N_C}$ and add $C \sqsubseteq Z$ to $O$;

(4) For each instance of $X \sqsubseteq Y$, if A occurs exactly once in it and a concept of the form $\exists R.C$ is present at the surface level of the left-hand side $X$, where $R$ is a role — and provided that $C$ contains A but is not in the form of $A \sqcap E$ as specified by A-NF V or VI, replace $C$ by a fresh definer $Z \in \mathsf{N_C}$ and add $C \sqsubseteq Z$ to $O$;

(5) For each instance of $X \sqsubseteq Y$, if A occurs exactly once in it and a concept of the form $\exists R.C$ is present at the surface level of the right-hand side $Y$, where $R$ is a role — and provided that $C$ contains A but is not in the form of $A \sqcap D$ as specified by A-NF II or III, replace $C$ by a fresh definer $Z \in \mathsf{N_C}$ and add $C \sqsubseteq Z$ to $O$;

The so-called *definers* [23], denoted as $Z$ in the above context, are newly-introduced concept names that serve as "abbreviations" for compound concepts when applying the normalization rules.

**LEMMA 1.** *Let $O$ be an arbitrary $\mathcal{ELI}$ ontology. Then $O$ can be transformed into A-normal form $O'$ by a linear number of applications of the normalization rule $1 - 5$. In addition, the size of the resulting ontology $O'$ is linear in the size of $O$.*

**LEMMA 2.** *Let $O$ be an arbitrary $\mathcal{ELI}$ ontology and $O'$ the normalized one obtained from $O$ using the normalization rules $(1) - (5)$. Then we have*

$$O \models C \sqsubseteq D \text{ iff } O' \models C \sqsubseteq D,$$

*for any $\mathcal{ELI}$-concepts $C$ and $D$ with $\mathsf{sig}(C \sqsubseteq D) \subseteq \mathsf{sig}(O)$.*

Lemma 1 states the termination and completeness of the normalization and Lemma 2 states its soundness.

### 3.2 R-Normal Form (R-NF)

**DEFINITION 3 (R-NORMAL FORM).** *We say that an axiom is in R-normal form (or R-NF) if it has one of the following forms, where*

|  | *R-NF* |  | *R-NF* |
|---|---|---|---|
| *I* | $C \sqsubseteq \exists r.D$ | *III* | $E \sqcap \exists r.F \sqsubseteq B$ |
| *II* | $C \sqsubseteq \exists r^-.D$ | *IV* | $E \sqcap \exists r^-.F \sqsubseteq B$ |

*(i) $r \in \mathsf{N_R}$, and (ii) B, C, D, E, and F are concepts that do not contain r. An $\mathcal{ELI}$-ontology $O$ is in R-NF if every r-axiom in $O$ is in R-NF.*

One can compute the R-NF of a given $\mathcal{ELI}$-ontology $O$ using a slightly adjusted approach for A-NF transformation.

# 4 DEFINER INTRODUCTION STRATEGY

Compared to the SOTA methods LETHE and FAME, our forgetting method introduces a novel normal form specification and exploits a non-traditional, yet notably cost-effective, definer introduction strategy for normalization, a factor that significantly bolsters the method's efficiency.

In closely examining LETHE and FAME, we delve into the intricacies of how they employ definers to facilitate the normalization of ontologies intended for the subsequent application of their respective forgetting rules. LETHE and FAME work on clauses of the form $L_1 \sqcup \ldots \sqcup L_k$, where each $L_i$ ($1 \le i \le k$) is a TBox literal, defined as:

$$A \mid \neg A \mid \exists r.Z \mid \exists r^-.Z \mid \forall r.Z \mid \forall r^-.Z,$$

where $r \in N_R$ and $A, Z \in N_C$. A salient observation is that LETHE mandates every $Z$ (essentially, any subconcept immediately below a $\exists$- or $\forall$-restriction) to be a definer at any stage of the forgetting process. In contrast, our method allows for a more flexible specification of $Z$. Such differentiation profoundly impacts the general applicability of the forgetting rules these methods employ, and by extension, has implications for their inferential efficiency.

For a deeper algorithmic understanding of LETHE's definer introduction strategy, we first fix some notations. By $\text{sig}_D(O)$ we denote the set of definers introduced in $O$, by $\text{Sub}^\forall_\exists(O)$ the set of all subconcepts of the form $\exists r^{(-)}.X$ or $\forall r^{(-)}.X$ in $O$, where $r \in N_R$ and $X$ is an arbitrary concept, and by $\text{Sub}_X(O)$ the set of all subconcepts $X$ present in $O$ with $\exists r^{(-)}.X \in \text{Sub}^\forall_\exists(O)$ or $\forall r^{(-)}.X \in \text{Sub}^\forall_\exists(O)$.

In the LETHE framework, which exploits a definer reuse strategy (i.e., LETHE consistently uses a definer to refer to identical subconcepts), an injective function $f$ can be defined over $\text{sig}_D(O)$, namely $f : \text{sig}_D(O) \to \text{Sub}_X(O)$. $f$ also exhibits surjectivity, given LETHE's exhaustive manner to introduce defines — LETHE mandates every subconcept immediately below an $\exists$- or $\forall$-restriction to be a definer. On the other hand, within our framework, $f$ is defined as non-surjective. However, for both methods, the number of definers — denoted as $|\text{sig}_D(O)|$ — relevant to the normalization of $O$, is bounded by $O(n)$. Here, $n$ corresponds to the count of $\exists$- and $\forall$-restrictions present in $O$. This implies a linear growth in the introduction of definers.

In the LETHE framework, a saturation-based reasoning procedure is employed to forget individual concept names from $O$. This is achieved by deriving new entailments and adding them to $O$ using a generalized resolution-based calculus called Res, as outlined by [7]. Notably, Res operates on the clausal normal form, as defined earlier in this section. LETHE uses a two-stage approach to iteratively compute the normal form of $O$. The first stage, termed as the *pre-resolution stage* (as previously detailed), witnesses LETHE's inaugural computation of $O$'s normal form to fire up Res. During this stage, definers are introduced in a *linear* and *static* manner, as previously discussed.

Transitioning to the subsequent *intra-resolution stage*, LETHE applies exhaustively Res's inference rules on $O$ until saturation is achieved at $\text{Res}(O)$, where new entailments are propagated from existing ones. For example, applying the $\forall\exists$-*role propagation* rule to $C_1 \sqcup \forall r.D_1$ and $C_2 \sqcup \exists r.D_2$ yields a new entailment $C_1 \sqcup C_2 \sqcup \exists r.(D_1 \sqcap D_2)$, for which LETHE has to introduce a fresh definer $D_{12} \in N_C$ to replace $D_1 \sqcap D_2$ for normalization, and then

adds $\neg D_{12} \sqcup (D_1 \sqcap D_2)$ to $O$ — definers are *dynamically* introduced as Res iterates over $O$. Following this process, an additional injective yet non-surjective function $f'$ emerges over $\text{sig}_D(\text{Res}(O))$, defined as $f : \text{sig}_D(\text{Res}(O)) \to \text{Sub}_X(\text{Res}(O))$, and the size of the codomain $|\text{Sub}_X(\text{Res}(O))| = 2^{|\text{Sub}_D(O)|}$. The number of the definers for the normalization of $\text{Res}(O)$ is bounded by $O(2^n)$, where $n$ is the number of $\exists$- and $\forall$-restrictions in $O$. Thus, LETHE introduces definers at an exponential rate during the intra-resolution stage. In contrast, our forgetting method restricts its normalization endeavors within the pre-resolution stage, indicating a linear trajectory in the introduction of definers during its entire forgetting span.

Definers are extraneous to the desired signature and thus should be excluded from the result of forgetting $\mathcal{F}$ from $O$. Consequently, in the worst case, LETHE will be tasked with discarding as many as $(2^n) + |\mathcal{F}|$ names and executing Res for $O(2^n) + |\mathcal{F}|$ iterations to compute the forgetting result. In contrast, our method introduces a maximum of $n$ definers, and in the worst case, only needs to activate the forgetting calculus (described next) $n + |\mathcal{F}|$ times.

# 5 THE FORGETTING METHOD

## 5.1 Calculus for Forgetting A

Our forgetting method exploits a two-step calculus to forget a single concept name A from $O$:

**Step I:** computes the A-normal form of $O$ as described in the previous section;

**Step II:** forgets A by exhaustively applying the inference rules in Figure 1.

While LETHE and FAME work on the clausal representation of $O$, our method directly addresses GCIs.

At the core of the exhaustive application of the inference rules is the endeavor to reveal all logical entailments regarding $\text{sig}(O) \backslash \{A\}$, and subsequently incorporate these new entailments into $O$. This process continues until $O$ becomes saturated w.r.t. A. By definition, $O$ is said to be *saturated* w.r.t. A iff every entailment $\alpha$ of an inference with A $\notin \text{sig}(\alpha)$ is already in $O$, i.e., $\alpha \in O$, or is redundant w.r.t. $O$, i.e., $O \backslash \{\alpha\} \models \alpha$. Upon reaching this state of saturation, all A-axioms can be safely removed from $O$, resulting in the forgetting of A from $O$ eventually.

The process of revealing implicit entailments from $O$ involves the combination of positive occurrences (GCIs taking A-NF I, II, and III) with negative occurrences (GCIs taking A-NF IV, V, and VI) of A (i.e., resolution upon A). This results in six distinct combination scenarios, labeled as IR1, IR2, IR3, IR4, IR5, and IR6, as depicted in Figure 1. By the exhaustive application of these inference rules, followed by the removal of all A-axioms, the outcome is a refined ontology, denoted as $O^{-A}$, devoid of any traces of A.

EXAMPLE 1. Consider the following $\mathcal{ELI}$-ontology $O$:

$$\{1. \; E \sqsubseteq \exists r^-.(F \sqcap \exists t.A), \; 2. \; \exists t.A \sqcap \exists t^-.E \sqsubseteq D\}$$

Let $\mathcal{F} = \{A\}$. The first step is to compute the A-NF of $O$ by applying the normalization rules as described earlier, where $Z_1 \in N_D$ is a fresh definer:

$$\{3. \; E \sqsubseteq \exists r^-.Z_1, \; 5. \; Z_1 \sqsubseteq F, \; 6. \; Z_1 \sqsubseteq \exists t.A, \; 2. \; \exists t.A \sqcap \exists t^-.E \sqsubseteq D\}$$

The above ontology is now in A-normal form. The second step is to apply the inference rules in Figure 1. Applying Rule IR5 to Axioms

IR1. $C \sqsubseteq A, A \sqcap E \sqsubseteq F \implies C \sqcap E \sqsubseteq F$

IR2. $C \sqsubseteq A, \exists s.(A \sqcap E) \sqcap F \sqsubseteq G \implies \exists s.(C \sqcap E) \sqcap F \sqsubseteq G$

IR3. $C \sqsubseteq A, \exists s^-.(A \sqcap E) \sqcap F \sqsubseteq G \implies \exists s^-.(C \sqcap E) \sqcap F \sqsubseteq G$

IR4. $C \sqsubseteq \exists r.(A \sqcap D), A \sqcap E_1 \sqsubseteq F_1, \ldots, A \sqcap E_n \sqsubseteq F_n$
$\implies C \sqsubseteq \exists r.(F_1 \sqcap \ldots \sqcap F_n \sqcap D)$
provided that: $O \models A \sqcap D \sqsubseteq E_1 \sqcap \cdots \sqcap E_n$
$C \sqsubseteq \exists r.(A \sqcap D), A \sqcap E \sqsubseteq F \implies C \sqsubseteq \exists r.D$
provided that: $O \not\models A \sqcap D \sqsubseteq E$

IR5. $C \sqsubseteq \exists r.(A \sqcap D), \exists s.(A \sqcap E) \sqcap F \sqsubseteq B$
$\implies C \sqsubseteq \exists r.D, C \sqcap F \sqsubseteq B$
provided that: $O \models A \sqcap D \sqsubseteq E$ and $O \models r \sqsubseteq s$
$C \sqsubseteq \exists r.(A \sqcap D), \exists s.(A \sqcap E) \sqcap F \sqsubseteq B \implies C \sqsubseteq \exists r.D$
provided that: $O \not\models A \sqcap D \sqsubseteq E$ or $O \not\models r \sqsubseteq s$

IR6. $C \sqsubseteq \exists r.(A \sqcap D), \exists s^-.(A \sqcap E) \sqcap F \sqsubseteq B$
$\implies C \sqsubseteq \exists r.D, C \sqcap F \sqsubseteq B$
provided that: $O \models A \sqcap D \sqsubseteq E$ and $O \models r \sqsubseteq s^-$
$C \sqsubseteq \exists r.(A \sqcap D), \exists s^-.(A \sqcap E) \sqcap F \sqsubseteq B \implies C \sqsubseteq \exists r.D$
provided that: $O \not\models A \sqcap D \sqsubseteq E$ or $O \not\models r \sqsubseteq s^-$

IR7. $C \sqsubseteq \exists r^-.(A \sqcap D), A \sqcap E_1 \sqsubseteq F_1, \ldots, A \sqcap E_n \sqsubseteq F_n$
$\implies C \sqsubseteq \exists r^-.(F_1 \sqcap \ldots \sqcap F_n \sqcap D)$
provided that: $O \models A \sqcap D \sqsubseteq E_1 \sqcap \cdots \sqcap E_n$
$C \sqsubseteq \exists r^-.(A \sqcap D), A \sqcap E \sqsubseteq F \implies C \sqsubseteq \exists r^-.D$
provided that: $O \not\models A \sqcap D \sqsubseteq E$

IR8. $C \sqsubseteq \exists r^-.(A \sqcap D), \exists s.(A \sqcap E) \sqcap F \sqsubseteq B$
$\implies C \sqsubseteq \exists r^-.D, C \sqcap F \sqsubseteq B$
provided that: $O \models A \sqcap D \sqsubseteq E$ and $O \models r^- \sqsubseteq s$
$C \sqsubseteq \exists r^-.(A \sqcap D), \exists s.(A \sqcap E) \sqcap F \sqsubseteq B \implies C \sqsubseteq \exists r.D$
provided that: $O \not\models A \sqcap D \sqsubseteq E$ or $O \not\models r^- \sqsubseteq s$

IR9. $C \sqsubseteq \exists r^-.(A \sqcap D), \exists s^-.(A \sqcap E) \sqcap F \sqsubseteq B$
$\implies C \sqsubseteq \exists r^-.D, C \sqcap F \sqsubseteq B$
provided that: $O \models A \sqcap D \sqsubseteq E$ and $O \models r^- \sqsubseteq s^-$
$C \sqsubseteq \exists r^-.(A \sqcap D), \exists s^-.(A \sqcap E) \sqcap F \sqsubseteq B \implies C \sqsubseteq \exists r^-.D$
provided that: $O \not\models A \sqcap D \sqsubseteq E$ or $O \not\models r^- \sqsubseteq s^-$

**Figure 1: Inference rules for forgetting A**

2 and 6 gives:

$$\{3.\ E \sqsubseteq \exists r^-.Z_1,\ 5.\ Z_1 \sqsubseteq F,\ 7.\ Z_1 \sqcap \exists t^-.E \sqsubseteq D,\ 8.\ Z_1 \sqsubseteq \exists t.\top\}$$

Definers are treated as regular concept names, and are eliminated once the names in $\mathcal{F}$ have been eliminated. Applying Rule IR7 to Axioms 3 and 7 gives $\{9.\ E \sqsubseteq \exists r^-.\top\}$. Applying Rule IR7 to Axioms 3, 5 and 8 gives $\{10.\ E \sqsubseteq \exists r^-.(F \sqcap \exists t.\top)\}$. Axiom 9 is redundant w.r.t. Axiom 10 and thus removed. Our method implements a set of straightforward simplifications. In this case, $\{10.\ E \sqsubseteq \exists r^-.(F \sqcap \exists t.\top)\}$ is a $\Sigma$-uniform interpolant of $O$, where $\Sigma = sig(O)\backslash\mathcal{F}$.

An external DL reasoner is utilized to check the side conditions of the inference rules. It is known that checking subsumption in $\mathcal{ELI}$ is ExpTime-complete [2].

LEMMA 3. *Let $O$ be an $\mathcal{ELI}$-ontology in A-NF, and $O^{-A}$ an ontology obtained from forgetting $\{A\}$ from $O$ using the inference rules in Figure 1, then we have:*

$$O \models C \sqsubseteq D \text{ iff } O^{-A} \models C \sqsubseteq D,$$

*for any $\mathcal{ELI}$-GCI $C \sqsubseteq D$ with $sig(C \sqsubseteq D) \subseteq sig(O)\backslash\{A\}$.*

Lemma 3 establishes the partial soundness of the calculus. Specifically, the derived ontology $O^{-A}$ fulfills the second condition necessary for it to be the result of forgetting $\{A\}$ from $O$. However, $O^{-A}$ may include definers which fall outside the scope of $sig(O)\backslash\{A\}$, potentially failing to fulfill the first condition. A discussion on this will follow shortly.

## 5.2 Calculus for Forgetting r

The calculus for role forgetting parallels that for concept forgetting. Specifically, the calculus proceeds in two steps — Step (1) computes the r-NF of $O$ as described in the previous section, and Step (2) forgets r by exhaustive application of the inference rules in Figure 2 to the normalized $O$.

IR10. $C \sqsubseteq \exists \mathsf{r}.D, F \sqcap \exists \mathsf{r}.E \sqsubseteq G \implies F \sqcap C \sqsubseteq G$
provided that: $O \models \exists \mathsf{r}.D \sqsubseteq \exists \mathsf{r}.E$

IR11. $C \sqsubseteq \exists \mathsf{r}.D, F \sqcap \exists r^-.E \sqsubseteq G \implies F \sqcap C \sqsubseteq G$
provided that: $O \models \exists \mathsf{r}.D \sqsubseteq \exists r^-.E$

IR12. $C \sqsubseteq \exists r^-.D, F \sqcap \exists \mathsf{r}.E \sqsubseteq G \implies F \sqcap C \sqsubseteq G$
provided that: $O \models \exists r^-.D \sqsubseteq \exists \mathsf{r}.E$

IR13. $C \sqsubseteq \exists r^-.D, F \sqcap \exists r^-.E \sqsubseteq G \implies F \sqcap C \sqsubseteq G$
provided that: $O \models \exists \mathsf{r}.D \sqsubseteq \exists r^-.E$

**Figure 2: Inference rule for forgetting r**

The process of eliciting implicit entailments from $O$ involves the combination of positive occurrences (GCIs taking r-NF I and II) with negative occurrences (GCIs taking r-NF III and IV) of r (i.e., resolving upon r). This results in four distinct combination scenarios, labeled as IR10, IR11, IR12, and IR13, as depicted in Figure 2. By the exhaustive application of these inference rules, followed by the removal of all r-axioms, the outcome is a refined ontology, denoted as $O^{-r}$, devoid of any traces of r. Likewise, an auxiliary DL reasoner is employed during the forgetting process to check the side conditions of the inference rules.

LEMMA 4. *Let $O$ be an $\mathcal{ELI}$-ontology in r-NF, and $O^{-r}$ an ontology obtained from forgetting $\{r\}$ from $O$ using the inference rules in Figure 2, then we have:*

$$O \models C \sqsubseteq D \text{ iff } O^{-r} \models C \sqsubseteq D,$$

*for any $\mathcal{ELI}$-GCI $C \sqsubseteq D$ with $sig(C \sqsubseteq D) \subseteq sig(O)\backslash\{r\}$.*

## 5.3 Properties of the Method

Forgetting is not always successful for $\mathcal{ELI}$ when cyclic dependencies exist over the intended names to be forgotten [17]. In these situations, the inference process might fall into an endless loop, causing the forgetting process to never terminate. Take an example where we aim to forget A from an $\mathcal{ELI}$-ontology $\{A \sqsubseteq \exists r^-.A\}$, which exhibits cyclic behavior over A. The result is $\{D_1 \sqsubseteq \exists r^-.D_1\}$, with $D_1 \in \mathsf{N_D}$ as a fresh definer. If one tried to forget $D_1$ from this result, it would yield a GCI of the same structure, specifically $\{D_2 \sqsubseteq \exists r^-.D_2\}$, with $D_2 \in \mathsf{N_D}$ as a fresher definer. This would result in an endless introduction of definers. Our method guarantees the termination of the forgetting process by giving up forgetting $D_1$. Instead, it keeps the initial definer $D_1$ in the resulting ontology, declaring an unsuccessful forgetting attempt and highlighting the method's inherent limitation. While cyclic situations might be tackled using fixpoints [5], as shown by Lethe, mainstream reasoning tools and the OWL API do not support fixpoints. Hence, to ensure our method remains practical, we opt out of including them in our target language, choosing practicality over completeness.

THEOREM 1. *Given any $\mathcal{ELI}$-ontology $\mathcal{O}$ and any forgetting signature $\mathcal{F} \subseteq sig(\mathcal{O})$ as input, our forgetting method always terminates and returns an $\mathcal{ELI}$-ontology $\mathcal{V}$. If $\mathcal{V}$ does not contain any definers, then it is a result of forgetting $\mathcal{F}$ from $\mathcal{O}$.*

## 6 EXPERIMENTS

We have developed a prototype of our forgetting method in Java using the OWL API Version 5.1.7[1] To assess its practicality, we juxtaposed its performance with the state-of-the-art forgetting method, Lethe [20] using two large corpora of real-world ontologies.[2] The first corpus was created from a snapshot of the Oxford ISG Library,[3] aggregating diverse ontologies from a myriad of sources. The second corpus was derived from a March 2017 snapshot of the NCBO BioPortal [30], which features biomedical ontologies.

From the Oxford ISG snapshot, we cherry-picked 488 ontologies where the logical axiom (GCI) count did not exceed 10,000. We then excluded those ontologies lacking $\exists$-restrictions or inverse roles, or exhibiting cyclic dependencies. This left us with 177 ontologies. To refine further, we distilled the remaining ontologies down to their $\mathcal{ELI}$-fragments by omitting GCIs not expressible within $\mathcal{ELI}$. This process resulted in a 7.4% reduction in total GCIs.

To provide granular insights into the performance of our method across differently-sized Oxford-ISG ontologies, we partitioned these selections into three distinct categories:

- PART I: 115 ontologies with $10 \leq |Onto| < 1000$;
- PART II: 51 ontologies with $1000 \leq |Onto| < 4999$;
- PART III: 11 ontologies with $5000 \leq |Onto| < 10000$.

Implementing the same strategy for the BioPortal case, we amassed a collection of 76 ontologies and categorized them as:

- PART I: 38 ontologies with $10 \leq |Onto| < 1000$.
- PART II: 28 ontologies with $1000 \leq |Onto| < 4999$.
- PART III: 10 ontologies with $5000 \leq |Onto| < 10000$.

---

[1] http://owlcs.github.io/owlapi/

[2] Note that a comparative analysis with Nui and Fame was precluded due to accessibility issues during the period of our study (they remained inaccessible as of October 8, 2023).

[3] http://krr-nas.cs.ox.ac.uk/ontologies/lib/

Table 1: Experimental results over Oxford-ISG and BioPortal (Time: Time Consumption, Mem: Memory Consumption, SR: Success Rate, TR: Timeout Rate, RER: Runtime Error Rate)

| Oxford | % | Part | Time (sec.) | Mem (MB) | SR | TR | RER |
|---|---|---|---|---|---|---|---|
| Lethe | 0.1 | I | 4.55 | 37.06 | 92.07 | 4.45 | 3.48 |
| | | II | 9.92 | 58.76 | 86.57 | 11.19 | 2.24 |
| | | III | 14.72 | 88.54 | 77.73 | 22.27 | 0.00 |
| | 0.3 | I | 12.76 | 52.24 | 86.98 | 9.54 | 3.48 |
| | | II | 29.50 | 75.16 | 74.88 | 22.88 | 2.24 |
| | | III | 41.75 | 123.11 | 67.91 | 32.09 | 0.00 |
| | 0.5 | I | 14.81 | 71.36 | 79.80 | 16.72 | 3.48 |
| | | II | 43.41 | 134.65 | 70.16 | 27.60 | 2.24 |
| | | III | 74.22 | 189.13 | 63.45 | 36.55 | 0.00 |
| Proto | 0.1 | I | 0.18 | 24.76 | 100 | 0.00 | 0.00 |
| | | II | 0.49 | 38.78 | 100 | 0.00 | 0.00 |
| | | III | 0.87 | 59.45 | 100 | 0.00 | 0.00 |
| | 0.3 | I | 0.30 | 35.72 | 100 | 0.00 | 0.00 |
| | | II | 0.77 | 51.11 | 100 | 0.00 | 0.00 |
| | | III | 1.12 | 86.82 | 100 | 0.00 | 0.00 |
| | 0.5 | I | 0.81 | 48.36 | 100 | 0.00 | 0.00 |
| | | II | 1.41 | 91.65 | 100 | 0.00 | 0.00 |
| | | III | 1.62 | 130.50 | 100 | 0.00 | 0.00 |
| **BioPortal** | % | Part | Time (sec.) | Mem (MB) | SR | TR | RER |
| Lethe | 0.1 | I | 5.11 | 39.96 | 92.00 | 5.37 | 2.63 |
| | | II | 11.23 | 59.04 | 85.14 | 11.29 | 3.57 |
| | | III | 15.01 | 95.83 | 74.30 | 25.70 | 0.00 |
| | 0.3 | I | 14.19 | 53.26 | 83.29 | 14.08 | 2.63 |
| | | II | 32.16 | 88.33 | 73.11 | 23.32 | 3.57 |
| | | III | 46.15 | 133.20 | 65.50 | 34.50 | 0.00 |
| | 0.5 | I | 14.34 | 76.48 | 77.24 | 20.13 | 2.63 |
| | | II | 45.98 | 140.11 | 69.00 | 27.43 | 3.57 |
| | | III | 81.81 | 187.93 | 60.60 | 39.40 | 0.00 |
| Proto | 0.1 | I | 0.17 | 21.45 | 100 | 0.00 | 0.00 |
| | | II | 0.45 | 34.11 | 100 | 0.00 | 0.00 |
| | | III | 0.85 | 52.16 | 100 | 0.00 | 0.00 |
| | 0.3 | I | 0.32 | 31.34 | 100 | 0.00 | 0.00 |
| | | II | 0.69 | 47.66 | 100 | 0.00 | 0.00 |
| | | III | 1.06 | 78.38 | 100 | 0.00 | 0.00 |
| | 0.5 | I | 0.77 | 44.16 | 100 | 0.00 | 0.00 |
| | | II | 1.36 | 88.67 | 100 | 0.00 | 0.00 |
| | | III | 1.55 | 120.94 | 100 | 0.00 | 0.00 |

A comprehensive breakdown of the refined ontologies from both sources can be found in the long version of this paper.

We designed three sets of experiments, targeting the forgetting of either 10%, 30%, or 50% of the concept and role names present within the signature of each ontology. These configurations align with well-established practices in the evaluation of forgetting methods, as evidenced in literature sources such as [20, 22, 46, 48]. For the selection of $\mathcal{F}$, we employed a shuffling algorithm to ensure a randomized choice. The experimental set-up involved a laptop equipped with an Intel Core i7-9750H processor, boasting 6 cores that peak at 2.70 GHz, and bolstered by 12 GB of DDR4-1600 MHz RAM. To ensure consistent performance metrics, we imposed constraints: a maximum run time of 300 seconds and an upper heap space limit of 9GB. We deemed a forgetting experiment successful if it met the following criteria:

(1) successful elimination of all names specified in $\mathcal{F}$.
(2) absence of any definers in the forgetting results, should they have been introduced during the process.
(3) completion within the stipulated 300-second window.
(4) operation within the set 9GB space limit.

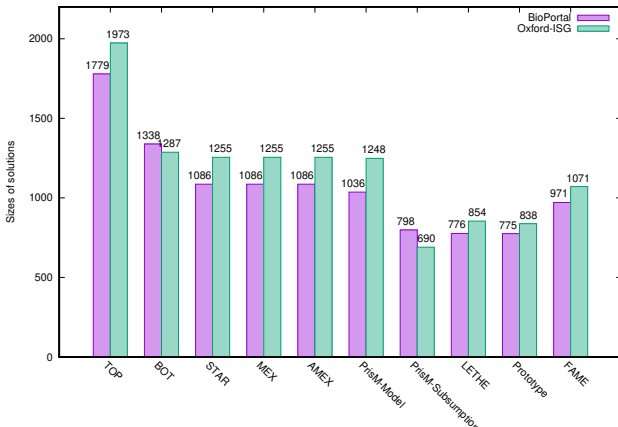

**Figure 3: Average |Onto| in output ontologies**

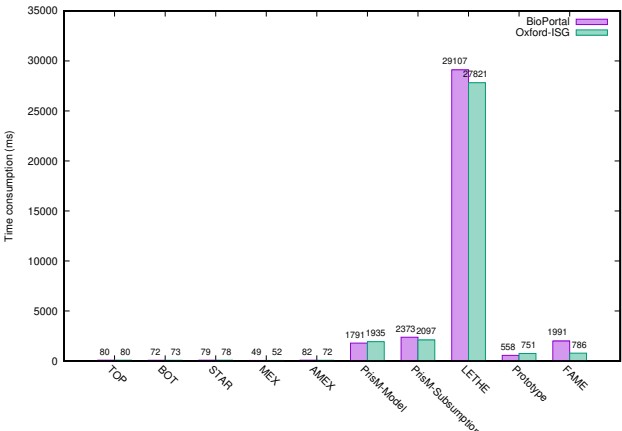

**Figure 5: Computation time consumption**

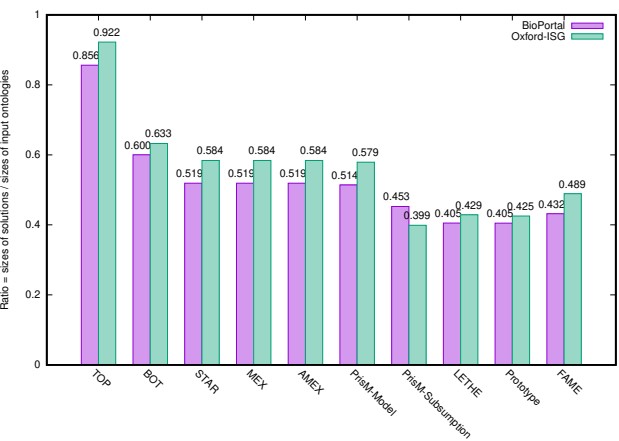

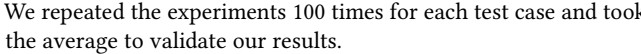

**Figure 4: Average ratio of |Onto|: input vs. output**

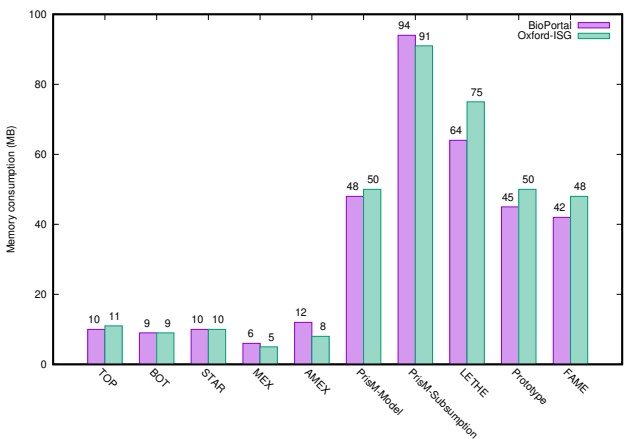

**Figure 6: Memory consumption**

We repeated the experiments 100 times for each test case and took the average to validate our results.

The results of our experiments are shown in Table 1. A notable observation is that our prototype registered a success rate of 100% across all evaluation tracks. A large portion of Lethe's failures was due to the timeout. Our logging of GPU and memory usage for each forgetting task indicated that Lethe generally demanded more computational resources compared to our prototype. Additionally, there was a stark contrast in processing times: our prototype consistently outperformed Lethe, being approximately 52 times faster on the Oxford-ISG dataset and 37 times faster on BioPortal. This marked increase in speed could be attributed to the distinct normalization approaches used by the two tools. A deeper dive into these results will follow later. Interestingly, Lethe incurred runtime errors when dealing with certain ontologies, but our prototype operated without any hitches, most likely because of the incompatibility between the OWL API version Lethe used and those ontologies.

To provide readers with a deeper insight into the inherent properties of our method and the nature of its forgetting results, we further conducted a comprehensive comparison with various types of modularization methods prevalent in the field, focusing on metrics such as result size, computation time, and memory consumption. A number of modularization approaches exist for creating views, each with unique properties and complexities. Among them, the MEX tool [18] extracts minimal modules from acyclic $\mathcal{ELI}$ ontologies with polynomial time complexity, while tools like AMEX [8] and PrisM [36] target other types of ontologies, with PrisM offering six inseparability notions. Locality-based methods and their extensions [11], such as TOP, BOT, and STAR, provide additional modularization strategies. With only MEX ensuring minimal modules, other tools generally approximate them.

Surprisingly, as a forgetting method, our prototype's output did not reflect the theoretical projections of an exponential size increase compared to the input ontologies. On the contrary, the forgetting results showcased impressive compactness in comparison to the

**Table 2: Definers introduced during forgetting (Oxford)**

| Onto Code | LETHE (0.1) | Proto (0.1) | LETHE (0.3) | Proto (0.3) | LETHE (0.5) | Proto (0.5) |
|---|---|---|---|---|---|---|
| 00646 | 2072 | 506 | 1329 | 0 | 2045 | 35 |
| 00645 | 1686 | 427 | 1005 | 0 | 1666 | 39 |
| 00522 | 884 | 359 | 2207 | 0 | 4233 | 266 |
| 00669 | 2628 | 1853 | 2960 | 638 | 2596 | 0 |
| 00696 | 1200 | 104 | 2992 | 133 | 6207 | 500 |
| 00523 | 4199 | 1361 | 2248 | 0 | 3173 | 20 |
| 00544 | 1349 | 8 | 5827 | 0 | 5885 | 0 |
| 00578 | 157 | 8 | 602 | 255 | 484 | 0 |
| 00356 | 1719 | 914 | 1719 | 228 | 1289 | 0 |
| 00367 | 23 | 5 | 24 | 0 | 31 | 0 |
| 00464 | 125 | 412 | 113 | 0 | 229 | 57 |
| 00513 | 14 | 29 | 37 | 4 | 28 | 0 |
| 00451 | 465 | 0 | 1156 | 0 | 2257 | 109 |
| 00445 | 27 | 0 | 120 | 5 | 88 | 0 |
| 00690 | 308 | 0 | 838 | 112 | 1276 | 175 |
| 00519 | 8 | 0 | 46 | 0 | 71 | 0 |
| 00527 | 97 | 0 | 501 | 57 | 371 | 0 |
| 00452 | 690 | 0 | 3072 | 257 | 2379 | 0 |
| 00650 | 111 | 0 | 353 | 0 | 663 | 49 |
| 00640 | 77 | 0 | 220 | 0 | 306 | 0 |
| 00495 | 417 | 0 | 1255 | 0 | 1772 | 0 |
| 00457 | 20 | 0 | 33 | 0 | 80 | 2 |
| 00494 | 446 | 0 | 1222 | 0 | 2335 | 133 |
| 00694 | 823 | 0 | 4580 | 480 | 5202 | 515 |
| 00469 | 5 | 0 | 10 | 0 | 46 | 0 |
| 00468 | 2 | 0 | 0 | 0 | 3 | 0 |
| 00497 | 1576 | 0 | 4223 | 0 | 6185 | 0 |
| 00520 | 39 | 0 | 73 | 6 | 67 | 0 |
| 00547 | 351 | 0 | 1952 | 327 | 1592 | 0 |
| 00433 | 47 | 0 | 141 | 0 | 221 | 20 |
| 00546 | 346 | 0 | 1950 | 255 | 1494 | 0 |
| 00591 | 10 | 0 | 66 | 0 | 67 | 0 |
| 00593 | 28 | 0 | 116 | 0 | 152 | 0 |
| 00357 | 331 | 0 | 1794 | 616 | 1794 | 285 |
| 00627 | 101 | 0 | 385 | 2 | 508 | 0 |
| 00545 | 1099 | 0 | 6108 | 0 | 6201 | 0 |
| 00592 | 41 | 0 | 70 | 0 | 104 | 0 |
| 00596 | 94 | 0 | 176 | 0 | 177 | 0 |
| 00423 | 129 | 0 | 338 | 0 | 458 | 0 |
| 00594 | 23 | 0 | 169 | 0 | 131 | 0 |
| 00770 | 499 | 0 | 2258 | 197 | 1968 | 24 |
| 00412 | 194 | 0 | 919 | 168 | 919 | 45 |
| 00413 | 188 | 0 | 550 | 0 | 776 | 0 |
| 00639 | 73 | 0 | 158 | 0 | 261 | 0 |
| 00605 | 33 | 0 | 75 | 6 | 69 | 0 |
| 00571 | 6 | 0 | 23 | 0 | 31 | 0 |
| 00411 | 37 | 0 | 137 | 0 | 172 | 0 |
| 00606 | 35 | 0 | 48 | 0 | 84 | 5 |
| 00570 | 38 | 4 | 19 | 0 | 32 | 0 |
| 00548 | 21 | 0 | 57 | 2 | 69 | 2 |
| 00366 | 8 | 0 | 19 | 5 | 34 | 2 |
| 00563 | 14 | 0 | 39 | 0 | 75 | 0 |
| 00629 | 111 | 0 | 419 | 3 | 471 | 9 |
| 00359 | 95 | 0 | 303 | 0 | 405 | 0 |
| 00403 | 467 | 0 | 1200 | 0 | 2479 | 404 |
| 00402 | 268 | 0 | 1407 | 294 | 1217 | 0 |
| 00358 | 37 | 2 | 90 | 0 | 85 | 0 |
| 00600 | 19 | 0 | 169 | 0 | 104 | 0 |
| 00006 | 211 | 0 | 488 | 0 | 1075 | 52 |
| 00562 | 10 | 0 | 32 | 0 | 36 | 0 |
| 00589 | 37 | 0 | 112 | 0 | 112 | 0 |
| 00505 | 9 | 0 | 3 | 0 | 9 | 0 |
| 00667 | 203 | 0 | 1181 | 270 | 890 | 0 |
| 00458 | 12 | 0 | 48 | 0 | 75 | 3 |
| 00498 | 1662 | 0 | 8316 | 1578 | 8316 | 754 |
| 00649 | 112 | 0 | 364 | 0 | 663 | 35 |
| 00514 | 27 | 0 | 35 | 0 | 36 | 0 |
| 00689 | 235 | 0 | 858 | 70 | 989 | 99 |
| 00515 | 168 | 0 | 649 | 35 | 773 | 27 |

**Table 3: Definers introduced during forgetting (BioPortal)**

| Onto Name | LETHE (0.1) | Proto (0.1) | LETHE (0.3) | Proto (0.3) | LETHE (0.5) | Proto (0.5) |
|---|---|---|---|---|---|---|
| DUO | 1 | 0 | 3 | 0 | 3 | 0 |
| ARO | 1304 | 0 | 2559 | 0 | 3162 | 5 |
| PCAO | 15 | 0 | 24 | 0 | 45 | 0 |
| AMPHX | 190 | 0 | 831 | 0 | 1334 | 656 |
| FAO | 4 | 0 | 15 | 0 | 21 | 0 |
| DMTO | 287 | 0 | 1307 | 0 | 1244 | 0 |
| HIO | 6 | 0 | 38 | 0 | 47 | 0 |
| HSAPDV | 75 | 0 | 475 | 388 | 421 | 16 |
| LMHA | 70 | 0 | 254 | 0 | 486 | 51 |
| PMDO | 2 | 0 | 12 | 0 | 16 | 0 |
| MMUSDV | 122 | 29 | 92 | 0 | 173 | 3 |
| HANCESTRO | 44 | 0 | 118 | 0 | 178 | 0 |
| EMAPA | 4088 | 0 | 9295 | 0 | 18035 | 2302 |
| PREO | 5 | 0 | 17 | 0 | 22 | 0 |
| AEO | 18 | 0 | 46 | 0 | 101 | 0 |
| EOL | 1 | 0 | 1 | 0 | 1 | 0 |
| LHN | 30 | 4 | 34 | 3 | 34 | 0 |
| ORDO | 15892 | 0 | 35407 | 78 | 40911 | 1898 |
| ORNASEQ | 3 | 0 | 2 | 0 | 5 | 0 |
| COGAT | 78 | 0 | 272 | 0 | 376 | 0 |

input ontologies, surpassing even the results produced by modularization techniques; these modules are mere syntactic subsets of the input ontologies; see Figures 3 and 4.

Regarding time consumption (see Figure 5), our prototype, when performing the same forgetting tasks, outpaced LETHE by what one might hyperbolically describe as "light years". Its speed was on par with the modularization methods; however, it is well-known that the computational complexity of forgetting is in general notably greater than that of modularization [4, 43].

Regarding memory consumption (see Figure 6), forgetting typically required more memory during computation than modularization. Yet, when comparing different forgetting methods, FAME and our prototype distinctly stood out, requiring only 66% to 70% of the memory that LETHE demanded, showcasing their efficiency.

Tables 2 and 3 present a detailed account of the definers introduced by LETHE and our prototype across a number of forgetting tasks. It is evident that, for any given task, our prototype introduced a substantially fewer number of definers compared to LETHE. In fact, in the Oxford-ISG settings, LETHE necessitated the introduction of definers in 65.0% of the forgetting tasks, whereas, for our prototype, this figure stood at 19.1%. In the BioPortal cases, these figures dropped to 26.3% for LETHE and a mere 5.2% for our prototype.

## 7 CONCLUSION AND FUTURE WORK

This paper presents a highly optimized forgetting method to produce signature-restricted views of acyclic $\mathcal{ELI}$ ontologies. Its enhanced efficiency results from a refined approach that reduces the number of definers needed for normalization. Despite the inherent computational challenges of the task, empirical evaluation demonstrates its algorithmic ascendancy over state-of-the-art tools.

Our immediate next step for future work is to enhance our current method to accommodate ABoxes. In addition, we also consider an adaptation of the method to more expressive DLs, such as $\mathcal{ALC}$ and its major decidable extensions [37, 41].

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
