# OpenReview forum: "Efficient Computation of Signature-Restricted Views for Semantic Web Ontologies"
_ACM.org/TheWebConf/2024/Conference — TheWebConf24 Oral_

### Official Review · Reviewer_JSTK · 2023-11-08

**Novelty:** 6
**Technical Quality:** 6

**Review:**

This paper proposes a novel forgetting strategy to be used in the computation of uniform interpolants of ontologies (ELI Description Logic is supported). The strategy is based on some normalization rules and some inference rules. While the theoretical worst-case complexity of the novel approach is very hard, the novel prototype implementation performs well in practice. In fact, the empirical evaluation shows that the prototype outperforms a competitor (Lethe) and performs similarly as other ontology modularization tools.

This paper addresses an important research problem, is well motivated and presented, is interesting for the conference audience, is correctly compared with the related work, and provides significant results.

As a suggestion to improve the presentation, Section 1 concludes by stating that the "basilenes are Lethe and Fame", but most of the experiments in Section 6 (Reported in Tables 1-3) consider Lethe and not Fame. I guess that it has already been shown in the literature that Lethe outperforms Fame, but this should be clearly stated in the paper and supported by some reference.

As an idea for futue work, while the authors study forgetting 10, 30, or 50% if concept and role names, it would be interesting to study the effects of forgetting concept and roles separately.

Minor comments:

 - The link to the supplementary material (https://github.com/anonymous-airesearcher/
www2024) does not work.

 - In Lemma 1, "rule 1-5" -> "rules 1-5"

 - In Section 3.2, I am missing some Lemma stating the correctness of the transformation process.

 - In Table 1, I would group by "ontology problems to be solved" and not by software tools, so that the results of Lethe and the prototype can be compared more easily as they would be in consecutive rows. Furthermore, you could use bold to highlight the best results.

**Questions:**

None

**Reviewer Confidence:**

3: The reviewer is confident but not certain that the evaluation is correct

**Scope:**

4: The work is relevant to the Web and to the track, and is of broad interest to the community

---

### Official Review · Reviewer_4R4g · 2023-11-16

**Novelty:** 5
**Technical Quality:** 6

**Review:**

The paper deals with the extraction of modules from ontologies. A module is an ontology that reflects only the relevant knowledge of an ontology for a particular purpose. Modules play an important role for tailoring the knowledge of an ontology to the scope of a particular application, to explore part of the knowledge modeled in an ontology, or to focus on a piece of knowledge from an ontology for debugging purpose or for refining it.

More precisely, the paper proposes a method to extract semantic modules from an ontology expressed in the ELI description logic wrt a signature of relations of interest (atomic concepts and atomic roles). Semantic modules depart from syntactic modules that must be subsets of the original ontology. In particular, an ontology and a semantic module of it only need to entail the same axioms wrt a given signature.

The proposed method builds on the well-known forgetting technique to build a semantic module that corresponds to a uniform interpolant of an ELI ontology wrt a target signature: the ontology and module entail the same axioms for this signature, and the signature of the module is the target signature.

To achieve this, an ELI ontology is first put in normal form, on which forgetting rules are then applied. The proposed normal form of an ontology is linear in the size of the original ontology and preserves the entailment of axioms wrt the signature of the original ontology. The proposed forgetting rules lead to an ELI ontology that turns out to be uniform interpolant wrt the target signature if the original ontology has no cyclic dependencies between axioms.

The proposed method is experimentally evaluated many ontologies of varying sizes from Oxford-ISG and BioPortal, and compared with the state-of-the-art baseline LETHE.
Reported experiments show that the proposed method significantly improves the baseline, for the considered case of acyclic ELI ontologies.

The problem addressed in the paper is relevant to the Semantic and Knowledge track of WWW. The paper is also well-written and easy to read and to follow, at least for the readers that are familiar with the description logic literature.
The novelty and originality of the paper lie in the proposed normalization and forgetting rules, which are clever though not technically challenging, while its significance lies in the very good performance it achieves in practice on acyclic ELI ontologies.
However, although the technical results look plausible, the provided URL at which proofs of these results (and code, datasets,…) are supposed to be available leads to HTTP 404 error.
Finally, also the reported experiments show good performance of the proposed method wrt LETHE used as SOTA baseline, it must be noted that LETHE is not specific to acyclic ELI ontologies but accommodates to (much) more expressive ontologies.

*** AFTER REBUTTAL ***

I thank you for providing the proofs of your technical results.

**Questions:**

Can you provide a valid URL for the proofs of the results?

**Ethics Review Description:**

No issue

**Reviewer Confidence:**

3: The reviewer is confident but not certain that the evaluation is correct

**Scope:**

4: The work is relevant to the Web and to the track, and is of broad interest to the community

---

### Official Review · Reviewer_bhgs · 2023-11-23

**Novelty:** 6
**Technical Quality:** 6

**Review:**

This paper considers creating signature restricted views using a uniform interpolation approach. The authors show that with good normalization and inference strategies, uniform interpolants can be efficiently computed, with good success rates in relevant experiments.

Overall, a well-structured and presented paper focusing on an interesting topic. The paper nicely explains the author’s approach with sufficient quality; however, readability could be improved adding as well more examples. Furthermore, the evaluation could be improved. The presentation of the significance of the proposed solution could be also improved.

***1. Introduction.
“Such constraints can limit the reusability of ontologies in specific real-world scenarios”. It is not clear why this is a limiting factor. As DBs do, role-based access to the data can be enforced.

I also value the provision of the GitHub repository with the code and the test datasets. However, although the authors mention that there is a long version of the paper there I could not locate it.

***1.1 Related work
In the related work section modularization approaches (TOP, BOT and STAR) and other competitors used in the experimental section should be reported and contrasted with your approach.

***2. Preliminaries/3.Normalization
The is not a single example in the two sections, which would enhance the readability of the paper.

*** 6. Experiments
Results nicely demonstrate the superiority of the proposed solution over several baselines.

It is not clear why in the experiment presented in Table 1, FAME is not included.

Further, it is not crystal clear what we see in the experiment with the modularization methods, since they do slightly different things. For example, we see that MEX is faster, providing slightly bigger ontologies, ensuring minimal module size as well.

I’m also missing a discussion on the significance of the results we see in the experiments. We can see that eventually uniform interpolants can be computed efficiently despite the high complexity. What about moving however beyond ELI ontologies or considering A-boxes as well? A bit of discussion is missing here.

Following, the comments and additional explanations provided by the authors, i consider that these questions have been adequately addressed. I wish to thank the authors for being quite detailed in their responses,

**Questions:**

It is not clear why in the experiment presented in Table 1, FAME is not included.
Furthermore, it is not entirely clear what we see in the experiment with the modularization methods, since they do slightly different things. For example, we see that MEX is faster, providing slightly bigger ontologies, ensuring minimal module size as well.

**Reviewer Confidence:**

3: The reviewer is confident but not certain that the evaluation is correct

**Scope:**

3: The work is somewhat relevant to the Web and to the track, and is of narrow interest to a sub-community

---

### Official Review · Reviewer_11oC · 2023-11-23

**Novelty:** 6
**Technical Quality:** 6

**Review:**

The paper proposes a new method of forgetting for using to compute ontology views for EL-ontologies restricted to a certain subsignature.
The work is well-motivated, the presentation is, while complex, logically presented and justified and, as far as I can, tell technically sound and the results are quite good compared to LETHE.
At the core lies a new normalization.


The introduction to Sec 3 could say a bit more why the two separate normal forms are needed (one for forgetting classes and one for roles?). The readers has to infer that from Sec. 5. Also, a bit more detail on the intuition behind the normal forms (why do we want each axiom in one of these forms?) would provide more insights to the reader.


21: where you first mention ELI-ontologies it is confusing because the term has not been introduced yet. The same is true on p. 2, 2nd paragraph: ELI has not been introduced and it is not clear how it differs from EL ontologies, which we can fairly assume most people are familiar with.

67-70: the sentence is missing something

Sec 2, 1st line: "countably infinite sets", while technically ok, it is misguiding since we can safely assume that any ontology has a finite set of concepts and role names.

Sec 4: last paragraph before Sec 5: can you explain why definers should be excluded. Why do we need to introduce them at all?

I acknowledge and thank the authors' for their responses. I believe that they have a good plan to address the raised issues. I believe these changes would further improve an already great paper.

**Questions:**

Can you provide some insights into the design of these NFs?

**Reviewer Confidence:**

3: The reviewer is confident but not certain that the evaluation is correct

**Scope:**

4: The work is relevant to the Web and to the track, and is of broad interest to the community

---

### Official Review · Reviewer_qPaW · 2023-11-30

**Novelty:** 6
**Technical Quality:** 6

**Review:**

A uniform interpolation based forgetting technique is discussed for creating restricted views of ELI ontologies. With normalization and inference strategies as the optimization techniques, the proposed technique turns out to be efficient. In the evaluation, two benchmark datasets are used to show that the proposed approach is computationally faster than the state-of-the-art approaches.

Strengths
1) Although theoretically, the computation cost is high, the authors show that in practice with good optimizations, the proposed approach is very efficient.
2) Good evaluation

Weaknesses
1) The relevance of this work to the Web is not clear.
2) Certain aspects of the paper, including the choices made (discussed in the questions to authors) are not clear.

The URL in footnote 3 on page 6 does not seem to be working.

**Questions:**

1) The proposed approach uses ontologies available on the Web for the evaluation, but does it answer any specific Web related scientific research challenge?
2) Are the rules in Figures 1, 2 complete? Please include a high-level sketch of the proof. How are these rules derived?
3) Why pick ontologies that have less than 10000 axioms? Does the proposed approach work well on large ontologies?
4) Why is there so much difference in the computational cost between theory and practice w.r.t your approach (forgetting methods)?
5) In Tables 2, 3, what do 0.1, 0.3, and 0.5 indicate?

**Reviewer Confidence:**

2: The reviewer is willing to defend the evaluation, but it is likely that the reviewer did not understand parts of the paper

**Scope:**

2: The connection to the Web is incidental, e.g., use of Web data or API

---

### Decision · Program_Chairs · 2024-01-22

**Decision:**

Accept (Oral)

**Comment:**

The paper presents a novel approach to ontology modularization by creating restrictive views through the use of uniform interpolation and forgetting. The proposed approach is novel and advances the current state of the art.
 The paper discusses strategies for the efficient computation of uniform interplants for EL-ontologies, and compares the proposed approach with other state of the art modularization techniques.
 The contribution is significant, the efficient computation of ontology modules is fundamental for many ontology engineering activities, from ontology reuse to query answering.
 The reviews have provided a number of constructive comments to improve the readability of the paper, that the authors are already taking into consideration. I would encourage the authors to implement as many of the suggested changes as feasible.

 Pros
 - The paper is generally well written and its contribution is very relevant for the Semantic and Knowledge track of WWW.
 - Code, experiment materials and proofs have been provided
 - The proposed technique is simple, elegant and efficient

 Cons
 - More examples could improve the readability of the paper e.g. for the notion of normalisation
 - The experiments and state of the art should clearly compare the proposed approach with existing ones (e.g. TOP, BOT and STAR)